# Total Phenolic Fraction (TPF) from Extra Virgin Olive Oil: Induction of apoptotic-like cell death in *Leishmania* spp. promastigotes and *in vivo* potential of therapeutic immunomodulation

Kalliopi Karampetsou[1,2,☉], Olga S. Koutsoni[1☉], Georgia Gogou[1,2], Apostolis Angelis[2], Leandros-Alexios Skaltsounis[2], Eleni Dotsika[1]*

**1** Laboratory of Cellular Immunology, Department of Microbiology, Hellenic Pasteur Institute, Athens, Greece, **2** Division of Pharmacognosy and Natural Product Chemistry, Department of Pharmacy, National and Kapodistrian University of Athens, Athens, Greece

☉ These authors contributed equally to this work.
* e.dotsika@pasteur.gr

## Abstract

### Background

Leishmaniasis is a serious multifactorial parasitic disease with limited treatment options. Current chemotherapy is mainly consisted of drugs with serious drawbacks such as toxicity, variable efficacy and resistance. Alternative bioactive phytocompounds may provide a promising source for discovering new anti-leishmanial drugs. Extra Virgin Olive Oil (EVOO), a key-product in the Mediterranean diet, is rich in phenols which are associated with anti-inflammatory, anti-cancer and anti-microbial effects. In this study, we investigate the anti-leishmanial effect of Total Phenolic Fraction (TPF) derived from EVOO in both *in vitro* and *in vivo* systems by investigating the contributing mechanism of action.

### Methodology/Principal findings

We tested the ability of TPF to cause apoptotic-like programmed cell death in *L. infantum* and *L. major* exponential-phase promastigotes by evaluating several apoptotic indices, such as reduction of proliferation rate, sub-G0/G1 phase cell cycle arrest, phosphatidylserine externalization, mitochondrial transmembrane potential disruption and increased ROS production, by using flow cytometry and microscopy techniques. Moreover, we assessed the therapeutic effect of TPF in *L. major*-infected BALB/c mice by determining skin lesions, parasite burden in popliteal lymph nodes, *Leishmania*-specific antibodies and biomarkers of tissue site cellular immune response, five weeks post-treatment termination. Our results show that TPF triggers cell-cycle arrest and apoptotic-like changes in *Leishmania* spp. promastigotes. Moreover, TPF treatment induces significant reduction of parasite burden in draining lymph nodes together with an antibody profile indicative of the polarization of Th1/Th2 immune balance towards the protective Th1-type response, characterized by the presence of IFN-γ-producing CD4+ T-cells and increased Tbx21/GATA-3 gene expression ratio in splenocytes.

**Data Availability Statement:** All relevant data are within the manuscript and its Supporting information files.

**Funding:** ED was awarded with Award Numbers MIS5002486 and MIS5028091, under the Operational Strategic Reference Framework (NSFR 2014–2020) (http://www.gsrt.gr). KK was awarded with the PhD Fellowship of Hellenic Foundation for Research and Innovation (HFRI) (GA. no.6ΝΔΘ46ΨΖ2Ν-ΣΣΟ) (http://www.elidek.gr/). The funders had no role in study design, data collection and analysis, decision to publish, or preparation of the manuscript.

**Competing interests:** The authors have declared that no competing interests exist.

## Conclusions/Significance

TPF exhibits chemotherapeutic anti-leishmanial activity by inducing programmed cell death on cell-free promastigotes and immunomodulatory properties that induce *in vivo* T cell-mediated responses towards the protective Th1 response in experimental cutaneous leishmaniasis. These findings enable deeper understanding of TPF's dual mode of action that encourages further studies.

## Author summary

Leishmaniasis is an important protozoan parasitic disease and the currently active pharmaceutical compounds used for its treatment are limited with various drawbacks. Therefore, the need for new drug alternatives is evident and the development of novel anti-leishmanial agents based on natural products (NPs) is challenging. Olive oil (OO) is an integral part of the exquisite "Mediterranean diet", constituting a high-value nutritional element associated with the lower incidence of coronary heart diseases and neurological disorders. Various studies conducted thus far, including human, animal, and *in vitro* models, attribute these intriguing biological properties to its adequate fatty acid profile and phenolic composition. Compositional studies have identified a plethora of phenolic compounds, including phenolic alcohols, secoiridoid derivatives, phenolic acids, lignans and flavonoids. Some of the most important biophenols found in OO are hydroxytyrosol, tyrosol, and oleuropein and its derivatives, oleocanthal and oleacein. Our recent studies concern the evaluation of the anti-leishmanial properties of NPs derived from *Olea europaea L.* (Oleaceae) in *in vitro* and *in vivo* models of experimental leishmaniasis. We have previously revealed Total Phenolic Fraction (TPF), as a promising product with anti-leishmanial properties in *in vitro* and *in vivo* systems. This study validates TPF as a potent inhibitory factor against *Leishmania* spp., driving parasites into an apoptotic-like cell death. Its leishmanicidal activity is further established in the *in vivo* murine experimental model of cutaneous leishmaniasis, evaluating also its ability to promote the host's protective Th1-type immune response. This study suggests the potential use of TPF against leishmaniasis because of its dual action as chemotherapeutic compound that eliminates parasite alone and also as immunostimulator of host's immune system.

## Introduction

Leishmaniasis is the second, after malaria, devastating tropical parasitic disease exhibiting a wide range of manifestations from self-healing cutaneous lesions (cutaneous leishmaniasis, CL) to severe visceral organ damage (visceral leishmaniasis, VL). It is considered as a major global health problem as it affects 0.7–1 million people every year in 100 countries all over the world, and presents very high clinical importance as an opportunistic infection in HIV co-infection clinical cases [1]. The available therapeutic options for leishmaniasis rely on limited chemotherapeutic agents, most of which are administered by the parenteral route inducing severe side-effects, while the rates of treatment failure are high [2]. The pentavalent antimony compounds, firstly introduced in the 1940s, have been used as the first line drugs against cutaneous or visceral leishmaniasis, despite the long lasting parenteral route of administration and high toxicity [2]. Paromomycin is an aminoglycoside antibiotic, discovered in the 1960s, with

leishmanicidal activity, also administered by the parenteral route with varying efficacy in different *Leishmania* species [3,4]. Miltefosine is the only oral drug approved for leishmaniasis treatment since 2002 but the emergence of a relevant relapse rate possesses the ultimate use of miltefosine as an option mainly for combination therapy [5,6]. Additionally, even though its side effects are considered manageable, its teratogenic potential makes it prohibitive for administration to pregnant women [7]. Treatment with liposomal amphotericin B was firstly applied in 1990, and since then it has been increasingly used against leishmaniasis. Despite its highest therapeutic index compared to all other existing anti-leishmanial drugs, its high cost makes it prohibitive in several endemic regions. Today, there is an imperative need for new drugs to confront leishmaniasis that will be effective, safe and affordable.

Traditional herbal medicines are gaining increased scientific interest and the development of new anti-leishmanial drugs based on alternative agents with pharmacological properties in the field of natural products is challenging. At the same time, a plethora of today's drugs used for a variety of diseases are plant-derived [8]. Plants possess a large repertoire of secondary metabolites that display a wide variety of pharmacological activities and the use of such medicinal plants as an innovative approach for various diseases treatment is reinforced. *Olea europaea L.* (Oleaceae) is the most significant plant in the Mediterranean area since its products are the benchmark of "Mediterranean diet" that has a profound positive influence on health outcomes and increased lifespan [9]. The biological properties of secondary metabolites found in *Olea europaea L.* have been studied previously and numerous biological activities have been defined [10,11]. Virgin olive oil (VOO) contains approximately 36 phenolic compounds that attract major scientific interest due to their biological properties such as anti-oxidant, anti-inflammatory, anti-microbial, neuroprotective and anti-cancer effects [12]. Thus, *Olea europaea L.* natural products display a promising role as drug candidates against leishmaniasis.

We have previously reported that three major VOO phenolic compounds, oleuropein, hydroxytyrosol and tyrosol, demonstrate anti-leishmanial activity in *in vitro* and *in vivo* assays [13]. Moreover, we recently demonstrated that the Total Phenolic Fraction (TPF) derived from Extra VOO (EVOO), a crude extract enriched in the phenolic compounds hydroxytyrosol and tyrosol, has been found to exert anti-leishmanial and immunomodulatory activities in *in vitro* and *in vivo* systems [14]. These findings indicate that TPF can be a promising anti-leishmanial compound since ideal antiparasitic compounds should exert both antimicrobial and immunomodulatory properties to provoke the simultaneous inhibition of parasite growth as well as the induction of an effective cell-mediated immune response.

Programmed cell death possesses a functional role in the biology of unicellular *Trypanosomatid* protozoan parasites like *Leishmania* [15]. The existence and identification of programmed cell death pathways in these organisms is critical for parasite biology and parasite-host interactions and could serve as a basis for developing new anti-parasitic drugs that take advantage of these pathways [16]. Induction of programmed cell death in *Leishmania* spp. promastigotes and intervention in the normal cell cycle sequence are some mechanisms that have been previously reported to contribute to anti-leishmanial effects of various medicinal plants [17].

The present study was undertaken in order to delineate the actual mechanism of action of TPF that contributes to its leishmanicidal activity. Thus, we determined: (a) the type of cell death initiated in TPF-treated *Leishmania* spp. promastigotes, (b) the effect in cell cycle and its outcome at parasite's cellular morphology, and (c) the efficacy of TPF treatment to promote protective type of host's immune response in a murine experimental model of cutaneous leishmaniasis. Understanding the key features of the parasitic type of cell death and the evolution of adaptive immunity to *L. major* is essential in developing effective strategies to control leishmaniasis based on medicinal plant products capable to exert pharmacological and immunomodulatory properties.

## Methods

### Ethics statement

Reporting of animal experiments followed the ARRIVE guidelines and experimental protocol was approved by the Institutional Protocols Evaluation Committee according to PD 56/2013 as adoption of Directive 2010/63/EU. Protocol license was issued by the Official Veterinary Authorities of the Prefecture of Attica in compliance with the above legislation in force.

### Extraction and chemical characterization of TPF

The recovery of TPF from EVOO was carried out by Centrifugal Partition Extraction (CPE) technique, as previously described [18]. Additionally, its qualitative and quantitative determination was performed by high performance liquid chromatography with diode array detection (HPLC-DAD), using a Thermo Finnigan HPLC system (Thermo Finnigan, San Jose, CA) coupled with a Spectral System UV6000LP PDA detector [14]. The major phenolic compounds present in TPF were hydroxytyrosol (HT), tyrosol (TYR), oleacein (OLEA) and oleocanthal (OLEO) [14].

### Culture of *Leishmania* spp. promastigotes

Two *Leishmania* species, *L. infantum* (zymodeme GH8, strain MHOM/GR/2001/GH8) and *L. major* (zymodeme LV39, strain MRHO/SU/59/P), causative of visceral and cutaneous leishmaniasis respectively, were used. Promastigotes were cultured in tissue flasks (CellStar, Greiner Bio-one, Germany), at 26 ˚C in a refrigerated incubator (Sanyo incubator mir-253 Electronic Biomedical, Japan), as has been previously reported [19].

Soluble *Leishmania* Antigen (SLA) was derived from $10^9$ *L. major* stationary-phase promastigotes as has been previously described [20].

### Determination of the half-maximal inhibitory concentration (IC$_{50}$)

The half-maximal inhibitory concentration (IC$_{50}$) of TPF was determined for *L. infantum* and *L. major* promastigotes via the resazurin reduction assay, as we have previously described [14,21]. Briefly, various increasing concentrations of TPF (0–400 μg/ml) were added in triplicates, in separate 96-well flat bottom tissue culture plates containing the optimal concentration of *L. infantum* and *L. major* promastigotes. Plates were incubated in a 26 ˚C incubator for 60 h and the resazurin solution was added in each well at a final concentration of 20 μg/ml. Plates were further incubated in a 26 ˚C incubator until a fluorescent red color was observed in the triplicate of promastigotes cultured only in the presence of RPMI-1640 medium plus 10% v/v fetal bovine serum (negative control group). Optical density (OD) was determined with an absorbance microplate reader (excitation at 570 nm, reference filter at 630 nm). The correlation between promastigote density and OD values was assessed by linear regression analysis with estimation of the coefficient of determination ($R^2$). The estimated IC$_{50}$ values are presented in Table 1.

**Table 1. Half-maximal inhibitory concentrations (IC$_{50}$ values) of TPF against *L. infantum* and *L. major* promastigotes.**

|  | IC$_{50}$ (μg/ml) | |
|---|---|---|
|  | *L. infantum* promastigotes | *L. major* promastigotes |
| TPF | 335.4 ± 24.7 | 207 ± 44.2 |

### Effect of TPF on *Leishmania* spp. promastigote cultures

*L. infantum* and *L. major* early exponential-phase promastigotes were treated with TPF at concentrations corresponded to the $IC_{50}$ and 2 x $IC_{50}$ values. Negative and positive control groups were untreated and HePC ($IC_{50}$)-treated parasites, respectively. The assessment of promastigote growth and their rate of multiplication was monitored by differential counting of dead and live promastigotes at 24 h intervals for 3 consecutive days by using the Trypan blue exclusion dye (Sigma–Aldrich, USA) in a Malassez counting chamber (hemocytometer) under a binocular optical microscope (Olympus, BH, Japan). For this, 0.1 ml of the parasite culture was diluted 1/10 in 5% formaldehyde solution (ScharlauChemie S.A., Spain) in RPMI-1640 medium, to slow down promastigote movement and this was further diluted with Trypan blue solution (0.4 w/v in phosphate-buffered saline/PBS, pH = 7.2).

### Effect of TPF on *Leishmania* spp. promastigote proliferation determined by CFSE staining

*L. infantum* and *L. major* early exponential-phase promastigotes were labelled with 8 μM of CFSE solution (Abcam, Cambridge, UK) in RPMI-1640 medium, for 15 min at room temperature according to manufacturer's instructions. Labelling was quenched with an equal volume of complete RPMI-1640 medium [22]. Parasites were then washed twice in PBS (pH = 7.2) in order to remove unincorporated CFSE, cultured in 5 ml complete RPMI-1640 medium and exposed in TPF ($IC_{50}$ and 2 x $IC_{50}$) or HePC ($IC_{50}$) as standard anti-leishmanial drug. Subsequently, the parasites were maintained at 26 ˚C and the rate of multiplication was determined at 24 h intervals, for 3 consecutive days, as follows: $10^6$ parasites of each culture were fixed in 4% w/v paraformaldehyde solution in PBS and CFSE fluorescence intensity was assessed in FACSCalibur flow cytometer (Becton-Dickinson, San Jose, CA, USA) and acquired data were analyzed with FlowJo V.10.0.8 software (Becton-Dickinson, Ashland, OR). The fluorescence intensity is reduced by half after each cell division, as CFSE fluorescence is partitioned between the daughter cells.

### Detection of TPF induced-morphological changes by microscopy

The morphological alterations in TPF-treated *L. infantum* and *L. major* promastigotes were microscopically recorded. Briefly, exponential-phase promastigotes were incubated with TPF at concentrations corresponded to the $IC_{50}$ and 2 x $IC_{50}$, for 24 h. Untreated parasites were used as the negative control group while parasites treated with the standard anti-leishmanial drug miltefosine (HePC), at the corresponding $IC_{50}$ concentration, were used as positive control [14]. 5 x $10^5$ parasites/sample were fixed with 2% w/v paraformaldehyde, washed with PBS and attached to glass slides pre-coated with 1 mg/ml poly-L-lysine (Sigma-Aldrich, USA). Samples were run in duplicates. Half of them were permeabilized with 0.2% v/v Triton X-100 for 5 min and incubated with 50 μg/ml RNase A (Promega, Madison, WI, USA) and 10 μg/ml propidium iodide (PI) (Sigma-Aldrich, USA) for 2 min [23], while the rest of them were not permeabilized, nor incubated with RNase A and PI. Coverslips were mounted in 10 μl Mowiol 4–88 (25% v/v glycerol, 100 mM Tris-HCl, pH = 8.5, Merck, Darmstadt, Germany) on microscope slides, sealed with nail polish and stored at 4 ˚C. Images of parasite morphology were acquired with a 40x objective using the Olympus IX-81 Cell-R imaging system and images of fluorescence after PI staining were acquired with a 63x objective using the Leica confocal TCS-SP microscope of the Light Microscopy Unit of the Hellenic Pasteur Institute. At least 20 cells from three independent experiments were observed for each experimental group.

## Cell cycle profiling of *Leishmania* spp. promastigotes

Cell cycle analysis of treated and untreated *L. infantum* and *L. major* promastigotes was performed using flow cytometry. Exponential-phase promastigotes were treated with two concentrations of TPF ($IC_{50}$ and 2 x $IC_{50}$) for 24, 48 and 72 h. For the same incubation periods, *L. infantum* and *L. major* promastigotes were also treated with HePC ($IC_{50}$, positive control groups), while the negative control groups were consisted of untreated parasites. Then, $10^6$ parasites/group were removed and fixed in 100% ethanol (v/v) for 2 min, as previously described [24]. Fixed parasites were harvested by centrifugation at 1600 rpm for 10 min at 26 ˚C and washed twice with PBS. Cells were then re-suspended in 100 μg/ml RNAse A and 50 μg/ml PI and incubated for 1 min at room temperature. Fluorescence intensity of PI (20,000 promastigotes/group) was evaluated on FACSCalibur flow cytometer and acquired data were analyzed with FlowJo V.10.0.8 software.

## Determination of phosphatidylserine externalization

The externalization of phosphatidylserine (PS) in the outer membrane of TPF-treated *Leishmania spp.* promastigotes was measured by double staining with Annexin-V and PI and was detected by FACS. *L. infantum* and *L. major* exponential-phase promastigotes were treated with two concentrations of TPF ($IC_{50}$ and 2 x $IC_{50}$) for 24, 48 and 72 h. For the same incubation periods, promastigotes were also treated with HePC ($IC_{50}$, positive control groups), while the negative control groups were consisted of untreated parasites. Triton X-100 (0.2% v/v) was used as control for 100% membrane permeabilization and maximum fluorescence [24]. After treatment, $10^6$ parasites/group were washed twice with PBS and then re-suspended in Annexin-V binding buffer (10 mM HEPES/NaOH (pH = 7.4), 140 mMNaCl, 5 mM $CaCl_2$). $10^5$ promastigotes/group were then placed in separate eppendorf tubes and incubated with Annexin-V-Fluos (Roche) and PI for 15 min at room temperature in dark, according to manufacturer's instructions. Samples (20,000 promastigotes/group) were assessed on FACSCalibur and acquired data were analyzed with FlowJo V.10.0.8 software.

## Measurement of mitochondrial transmembrane potential ($\Delta\Psi m$)

Changes in mitochondrial transmembrane potential ($\Delta\Psi$m) were analyzed by FACS using the MitoProbe JC-1 assay kit (Molecular probes, US) [25]. JC-1 probe (5′,6,6′-tetrachloro-1,1′,3,3′-tetraethylbenzimidazolylcarbocyanine iodide) aggregates in mitochondria and fluoresces red at higher transmembrane potential but at lower transmembrane potentials it remains as monomers in the cytoplasm that fluoresces green. Consequently, mitochondrial depolarization is indicated by a decrease in the red/green fluorescence intensity ratio (FL-2/FL-1; 590 nm/530 nm). Briefly, exponential-phase promastigotes were incubated with TPF ($IC_{50}$ and 2 x $IC_{50}$) and HePC ($IC_{50}$) for 24, 48 and 72 h. Negative control groups were untreated parasites. Promastigotes were washed in PBS and re-suspended (1 x $10^6$/ml) in 1 ml of PBS containing JC-1 dye at a final concentration of 2 μM [26]. Parasites were incubated in the dark for 20 min at room temperature and washed twice in PBS to remove the non-internalized dye. Maximum depolarization was obtained in the presence of the mitochondrial uncoupler carbonyl cyanide 3-chlorophenylhydrazone (CCCP, 50μM) and unstained parasites were used to set background fluorescence. Samples (20,000 promastigotes/group) were assessed on FACSCalibur and acquired data were analyzed with FlowJo V.10.0.8 software.

## Measurement of intracellular reactive oxygen species (ROS) generation in *Leishmania* spp. promastigotes

Intracellular ROS generation in TPF-treated *Leishmania* spp. promastigotes was assessed by FACS with the use of $H_2DCFDA$ cell permeant probe (Life Technologies, NY, USA), as previously described [27]. *L. infantum* and *L. major* exponential-phase promastigotes were treated with $IC_{50}$ and 2 x $IC_{50}$ concentrations of TPF and the induction of ROS was estimated at 24, 48 and 72 h. More specifically, at the end of incubation periods, 5 x $10^6$ parasites/group were washed in PBS and re-suspended in PBS containing $H_2DCFDA$ (20 μM). Parasites were incubated for 20 min at room temperature in dark. Hydrogen peroxide ($H_2O_2$, 1 mM) was used as positive inducer of ROS for untreated parasites that were respectively incubated for 15 min at room temperature in dark. The cells were analyzed with FACSCalibur using FlowJo V.10.0.8 software and acquired data are expressed as the intensity of fluorescence (Geo Mean).

## Animals and experimental protocol

Age-matched (8-10-weeks old) female BALB/c mice (n = 15) were obtained from the breeding unit of Hellenic Pasteur Institute (HPI, Athens, Greece). All experimental animals were housed in a specific pathogen-free animal facility, at a temperature of 22–25 ˚C and a photoperiod of 12 h. Mice were subcutaneously infected with $10^6$ *L. major* stationary phase promastigotes. At 7 days post-infection, mice were randomly assigned in three groups. TPF was intraperitonealy administered at 20 mg/kg body weight (b.w.) every other day for 28 days. HePC, as reference drug, was administered by oral gavage at 15 mg/kg b.w for 23 consecutive days.

## Evaluation of parasite load in popliteal lymph nodes

The number of parasites in popliteal lymph nodes was quantified by limiting dilution assay, five weeks post-treatment termination. Briefly, popliteal lymph nodes were weighed before homogenization and then serially diluted in 96-well flat bottom microtiter plates (Sarstedt, Numbrecht, Germany) with Schneider's insect medium (Sigma, St. Louis, MO, USA) supplemented with 20% fetal bovine serum (FBS) (Gibco, Paisley, UK). The number of viable parasites per mg of tissue was determined from the highest dilution at which promastigotes could be grown after 7 days of incubation at 26 ˚C [28].

## Detection of *Leishmania*-specific IgG antibodies

Five weeks post-treatment termination, sera from treated and untreated mice were analyzed by enzyme-linked immunosorbent assay (ELISA) for the presence of IgG1 and IgG2a specific antibodies against soluble *Leishmania* total antigen (SLA). Briefly, 96-well microtiter plates (Sarstedt, Numbrecht, Germany) were coated with 5 μg/ml of SLA in carbonate buffer (15 mM $Na_2CO_3$, 35 mM $NaHCO_3$), pH 9.6 and left overnight at 4 ˚C. Then, plates were blocked with 2% bovine serum albumin (BSA) (AppliChem GmbH, Germany) and serial dilutions of serum samples (1/20 to 1/40960) were added. Then, plates were incubated with either biotin-labeled rat anti-mouse IgG1 (500 ng/ml) or IgG2a (250 ng/ml) (AbDSerotec, Oxford, UK) followed by the addition of horse radish peroxidase (HRP) streptavidin solution (1/5000, AbD-Serotec). Absorbance was read at 450 nm and results are expressed as antibody titer ± SD. Antibody titer was determined as the maximum dilution in which the sample's OD value was above the cut-off level (i.e. twice the mean value of the blank wells).

## Evaluation of cytokine production by intracellular staining

Mice from each experimental group were sacrificed five weeks post-treatment termination and spleens were aseptically removed and homogenized mechanically in complete RPMI-1640 culture medium. Spleen cells were seeded in 24-well flat bottom plates (Sarstedt, Numbrecht, Germany) at a density of $10^6$ cells/ml and stimulated with 25 μg/ml of SLA for 24 h, in 37 ˚C and 5% $CO_2$, for obtaining antigen specific responses. Subsequently, cells were incubated with 2.5 μg/ml of Brefeldin A (Fluka, Germany) for 4 h and then fixed with 2% w/v paraformalde-hyde (PF) solution in PBS for 15 min at room temperature, as previously described [29]. Cells were then permeabilized at room temperature for 5 min in permeabilization wash buffer (PBS + 3% FBS + 0.1% saponin) and double stained with fluorescein isothiocyanate (FITC)-conjugated anti-mouse CD4+ (clone RM4-5) (BDPharmingen) and phycoerythrin (PE)-conjugated anti-mouse IL-4 (11B11 clone) and IFN-γ (XMG1.2 clone) monoclonal antibodies (BDPharmingen) for 30 min at 4 ˚C according to manufacturer's instructions. Control unstained samples were also similarly processed for all the above cases. Thereafter, the cells were re-suspended in PBS, and 20,000 cells/sample were analyzed on FACSCalibur and acquired data were processed with FlowJoV.10.0.8 software. The first gate of spleen cells was set on forward (FSC) and side scatter (SSC) to exclude debris and dead cells and to select lymphocyte population. Leukocytes were then selected by gating on CD4+ cells and further plotted against IFN-γ and IL-4.

## Transcription factors gene expression analysis

Spleen cells were obtained from all experimental groups, five weeks post-treatment termination and total mRNA was isolated using an RNeasy Mini kit (Qiagen, Germany), according to manufacturer's instructions. The RNA concentration and purity were determined by measuring optical density (OD) on NanoDrop 2000 spectrophotometer (Thermo Scientific, USA). The amount of 1 μg of extracted RNA was used as template for cDNA synthesis, as previously described [30].

Real-time polymerase chain reaction (real-time PCR) was performed using an Exicycler 96 thermocycler (Bioneer, Korea), with a SYBR Green PCR Master Mix (KapaBiosystems, USA). Specific primers used to amplify three genes of interest: T-box transcription factor (Tbx21), trans-acting T-cell-specific transcription factor (GATA-3) and glyceraldehyde dehydrogenase of the 3-phosphatase (GAPDH) were designed by Qiagen (QuantiTect Primer Assays; Qiagen, Netherlands). All qPCR experiments were performed in three replicates and non-template controls and reverse transcription controls were additionally performed. The PCR was conducted according to Qiagen's PCR protocol for the QuantiTect Primer Assays and cycling conditions were as follows: 94 ˚C for 10 min, followed by 40 cycles at 94 ˚C for 10 s and 60 ˚C for 30 s. The expression level of GAPDH mRNA was used to normalize data of mRNA quantification while the average of healthy controls was used as calibrator. All expression levels were computed via the ΔΔCt method [31].

## Statistical analysis

Data are representative of at least three independent experiments and are presented as mean values ± SD. In the *in vivo* procedure five animals per group were used. Differences between means were analyzed for significance using the two-sided Mann-Whitney test (IBM SPSS Statistics software, version 24) and were considered significant at 0.05 level of confidence.

## Results

### Effect of TPF on *Leishmania* spp. promastigote viability

The promastigote viability assay was performed using the resazurin reagent which can be irreversibly reduced by enzymes in viable cells generating a red fluorescent resorufin product [32]. The TPF dose-dependent inhibitory activity against *Leishmania* spp. promastigotes was observed using concentrations up to 400 μg/ml as it is shown in Fig 1. The presence of TPF in promastigote cultures for 60 h, inhibited *L. infantum* and *L. major* promastigotes growth and the 50% inhibitory concentration was determined at 335.4 μg/ml for *L. infantum* and 207 μg/ml for *L. major*, respectively.

The viability of promastigotes was estimated with the resazurin-based assay. Various increasing concentrations of TPF (0–400 μg/ml) were added in tissue culture plates containing *L. infantum* (A) and *L. major* (B) promastigotes. Plates were incubated in a 26 ˚C incubator for 60 h and optical density was determined with an absorbance microplate reader with excitation at 570 nm and reference filter at 630 nm. Data are presented as mean values ± SD of three independent experiments. Symbol of * indicates statistically significant differences compared to control group (promastigotes non-exposed to TPF).

### TPF inhibits the proliferation of *Leishmania* spp. promastigotes

In order to evaluate the effect of TPF on parasite replication, early-exponential phase promastigotes were cultured in the presence of two increasing concentrations of TPF ($IC_{50}$ and 2 x $IC_{50}$). The quantitative determination of the multiplication rate of TPF-treated *L. infantum* and *L. major* promastigotes was achieved by measuring the cell density every 24 h, over the 72 h of culture by staining with Trypan blue and counting in a hemocytometer chamber (Fig 2). Untreated parasites exhibited an exponential pattern of growth, while TPF treatment resulted in a drastic reduction of promastigote replication. The starting population was 10 x $10^6$ parasites/ ml for both *Leishmania* species. As it is shown in Fig 2A, the number of *L. infantum* parasites was significantly reduced by 78%, 84% and 89% at 24, 48 and 72 h respectively in the presence of the $IC_{50}$ concentration of TPF (p values = 0.046, 0.050 and 0.046, respectively). The number of *L. infantum* parasites treated with the 2 x $IC_{50}$ concentration of TPF, exhibited a 68%, 86% and 85% reduction at 24, 48 and 72 h, respectively, compared to untreated parasites. It is noteworthy that TPF exhibited stronger effect in *L. infantum* growth than the

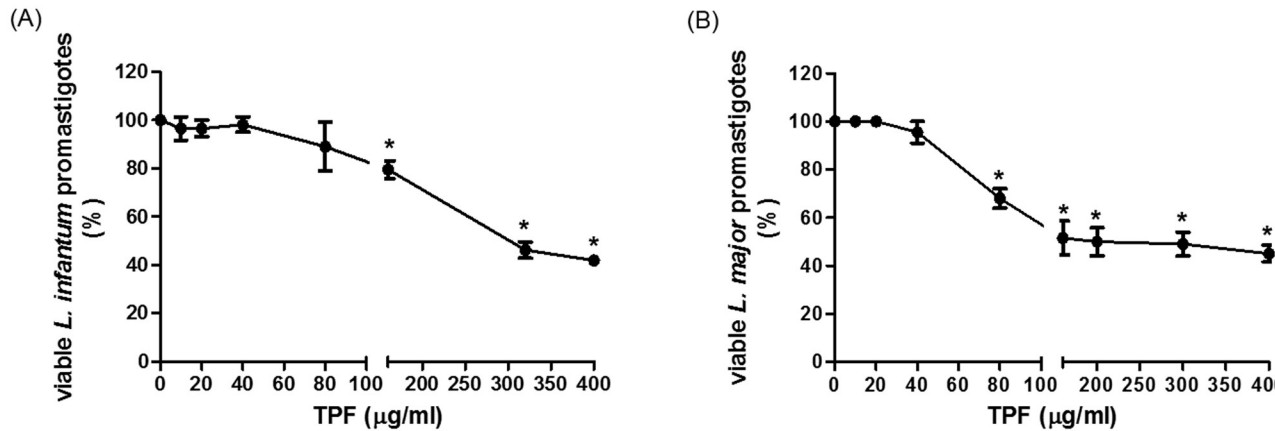

**Fig 1. Effect of TPF on the viability of *L. infantum* and *L. major* promastigotes.**

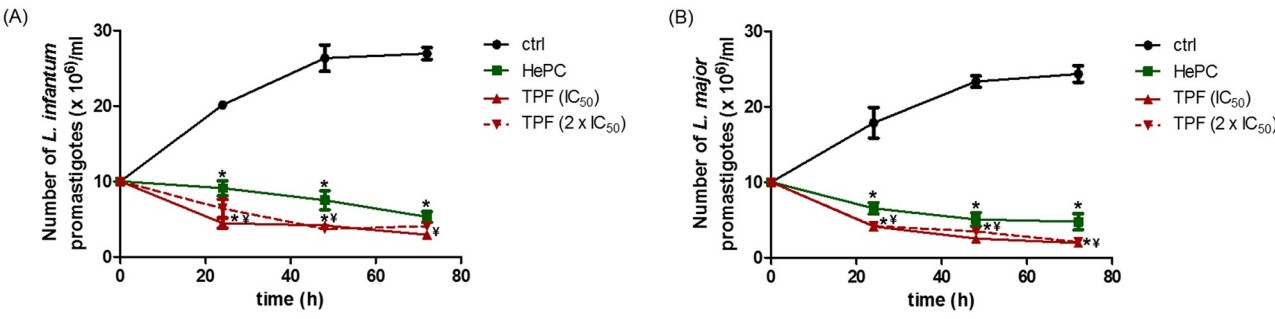

**Fig 2. Growth kinetics of *Leishmania* spp. promastigotes exposed to TPF.**

standard anti-leishmanial drug, HePC ($p \leq 0.050$). The corresponding population size of TPF ($IC_{50}$)-treated *L. major* promastigotes reached 4.1, 2.5 and 1.9 x $10^6$ parasites/ml over the 3-day culture period also indicating a marked decrease of parasite growth (77% to 92%) compared to untreated parasites, while the number of the 2 x $IC_{50}$-treated parasites exhibited an average 85% decrease over the 3-day culture period (Fig 2B). Moreover, the effects of both concentrations of TPF in *L. major* growth were more potent compared to effect of HePC ($p \leq 0.050$) (Fig 2B).

Furthermore, the qualitative determination of the promastigote proliferation under TPF exposure was determined with the use of the vital fluorescent stain CFSE. Cell divisions in each population were assessed by measuring daily the relative fluorescence intensity of cultures, during a 3-day culture period using flow cytometry. Discrete peaks and progressive decrease of CFSE fluorescence at different time intervals generally consist an indication for cell divisions. Fig 3A and 3B show the CFSE fluorescence intensity values for control, TPF-treated and HePC-treated *L. infantum* and *L. major* promastigotes, respectively. The fluorescence patterns observed in both *L. infantum* and *L. major* TPF-treated promastigotes indicated that treatment with TPF was more intensive than the reference drug HePC, also exhibiting a clear decelerated fluorescence decrease compared with untreated parasites. Moreover, the single parameter histogram overlays representative of one experiment are presented in S1 Fig.

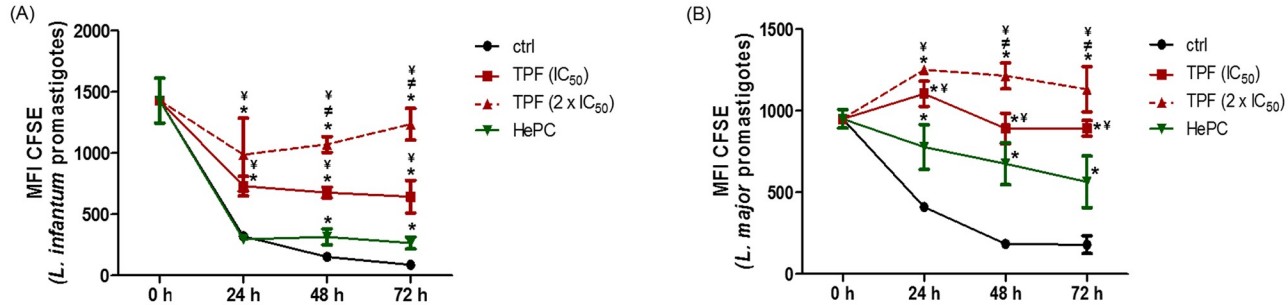

**Fig 3. *Leishmania* spp. promastigote proliferation determined by CFSE staining.** *L. infantum* (A) and *L. major* (B) early exponential-phase promastigotes were treated with $IC_{50}$ and 2 x $IC_{50}$ concentrations of TPF and their proliferation rate was qualitatively monitored at 24 h intervals for 3 consecutive days by CFSE staining and subsequent analysis of fluorescence intensity in FACS. HePC ($IC_{50}$)-treated and untreated parasites were used as positive and negative control groups, respectively. The results are expressed as mean fluorescence intensity values ± SD of three independent experiments. Symbols of $^*$ and $^¥$ indicate statistically significant differences compared to negative and positive control groups, respectively, while $^{\neq}$ indicates significant differences between TPF groups.

*L. infantum* (A) and *L. major* (B) early exponential-phase promastigotes were treated with $IC_{50}$ and 2 x $IC_{50}$ concentrations of TPF and their growth was quantitatively monitored at 24 h intervals for 3 consecutive days by differential counting of dead and live parasites using the Trypan blue exclusion dye. HePC ($IC_{50}$)-treated and untreated parasites were used as positive and negative control groups, respectively. Data are presented as mean values ± SD of three independent experiments. Symbols of [*] and [¥] indicate statistically significant differences compared to negative and positive control groups, respectively.

## TPF induces morphological changes in *Leishmania* spp. promastigotes

The cell changes at early apoptosis include membrane blebbing, cell shrinkage and pyknosis while apoptotic cells appear as round or oval mass with nuclear fragmentation and chromatin condensation [33,34]. We employed microscopy techniques in combination with PI nucleous staining in order to examine cellular morphological changes in *Leishmania* promastigotes after 24 h of exposure to TPF. Fig 4, shows the representative field of view of either untreated *L. infantum* (Fig 4A) and *L. major* (Fig 4B) promastigotes or treated with two concentrations of TPF, $IC_{50}$ and 2 x $IC_{50}$. While untreated promastigotes exhibited typical morphology of elongated cells with long flagellum, the morphology of most treated parasites was in contrast. TPF-treated parasites were round or ovoid shaped exhibiting higher fluorescence intensity, indicating DNA damage. HePC-treated promastigotes presented also remarkable shrinkage of body and were also intensively stained with the fluorescent dye.

Morphology of exponential-phase *L. infantum* (A) and *L. major* (B) promastigotes after treatment with TPF ($IC_{50}$ and 2 x $IC_{50}$) for 24 h. Images of parasite morphology were acquired with a 40x objective using the Olympus IX-81 Cell-R imaging system and images of fluorescence after PI staining were acquired with confocal fluorescence microscopy (TCS-SP, Leica). At least 20 cells from three independent experiments were observed from each experimental group.

## TPF induces sub-G0/G1 phase cell cycle arrest in *L. infantum* and *L. major* promastigotes

Cleavage of chromosomal DNA into oligonucleosomal size fragments is a biochemical hallmark of apoptosis. To verify whether TPF induces apoptotic-like modalities in L. infantum and L. major promastigotes, parasites were treated with two concentrations of TPF ($IC_{50}$ and 2 x $IC_{50}$) for 24, 48 and 72 h, and then were permeabilized and labeled with PI. Cell cycle progression was analyzed through flow cytometry. The amount of bound dye is correlated with the DNA content and the observation of a hypodiploid is translated into fluorescence intensity lower than that of G0/G1 cells, i.e. a 'sub-G0/G1' peak in the DNA histogram [35–37]. The results indicated that TPF induced sub-$G_0$ phase cell cycle arrest of both *L. infantum* and *L. major* promastigotes. Untreated *L. infantum* and *L. major* parasites had 0.7% and 1.2% of cells at sub-G0/G1 phase, respectively, at 24 h. Significant differences were observed in the relevant number of parasites for both tested concentrations of TPF. More specifically, we observed a 35-fold (24.5% of cells) and a 24-fold (17% of cells) increase in the sub-G0/G1 cells after a 24 h treatment period of *L. infantum* promastigotes with $IC_{50}$ and 2 x $IC_{50}$ concentrations of TPF, respectively (Fig 5A and 5C). Accordingly, *L. major* promastigotes also treated with $IC_{50}$ and 2 x $IC_{50}$ concentrations of TPF for 24 h, exhibited a 17- and a 20-fold increase (20% and 24% of cells, respectively) (Fig 5B and 5D). Noticeably, the effect of TPF at the $IC_{50}$ concentration was similar or even more potent compared to HePC which has been previously proven to cause sub-$G_0$ cell arrest in *Leishmania* promastigotes [38], for both species. The TPF-induced sub-G0/G1 phase cell cycle arrest of *Leishmania* spp. promastigotes was also verified by the

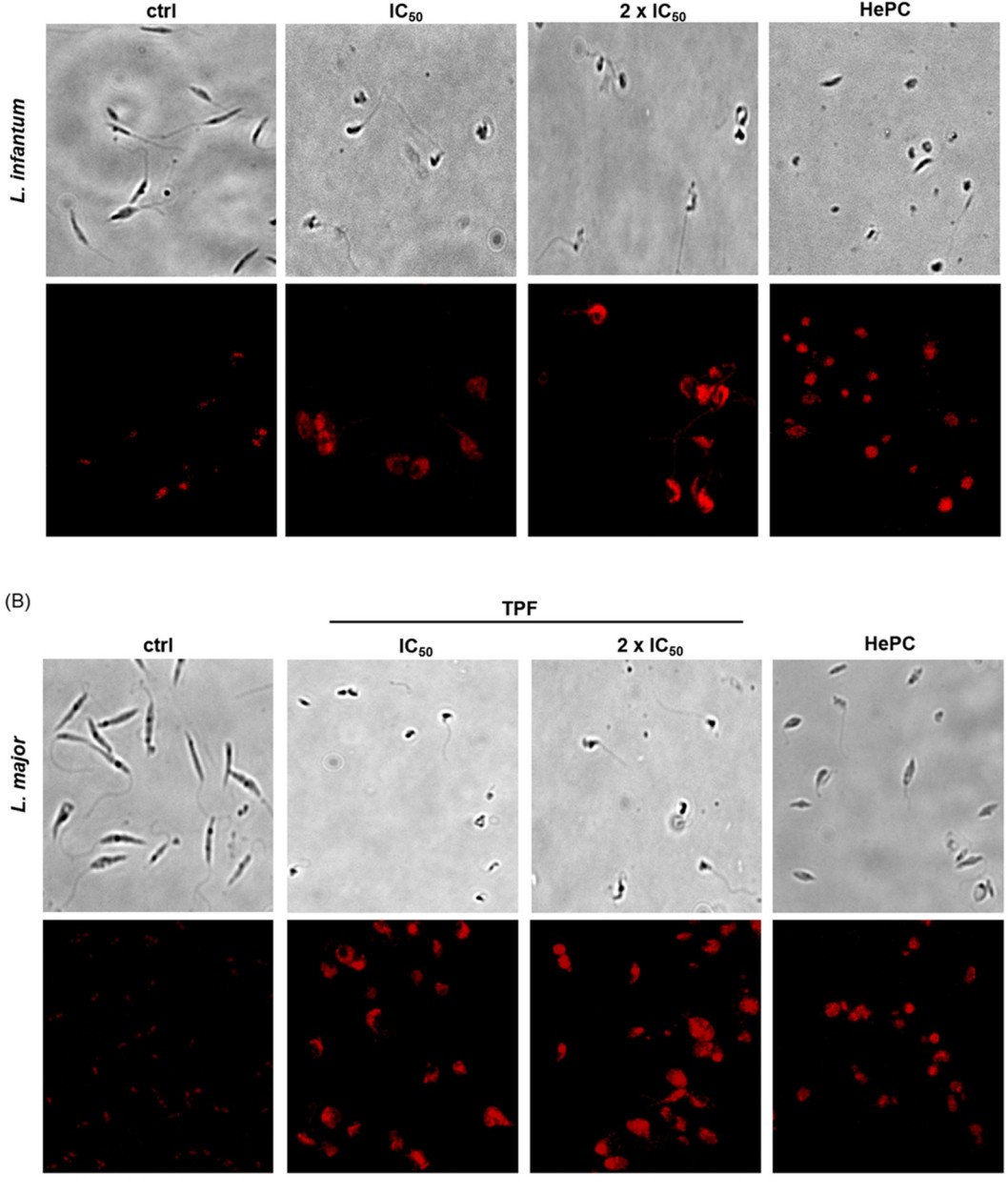

**Fig 4. Microscopy images of *Leishmania* spp. promastigotes.**

significant lower percentages of cells detected in the G1 peak region, compared with untreated parasites. Untreated *L. infantum* and *L. major* parasites had 74% and 59% of cells at G1 phase, respectively, at 24 h, while the relevant percentages of *L. infantum* promastigotes were 21% and 47% after treatment with $IC_{50}$ and 2 x $IC_{50}$ concentrations of TPF (Fig 5A and 5C). Accordingly, the percentages of *L. major* promastigotes in G1 peak region after exposure to both TPF concentrations, were 10% and 23%, respectively (Fig 5B and 5D). Moreover, after 48 and 72 h of treatment, the proportions of TPF-treated promastigotes in the G1 region were

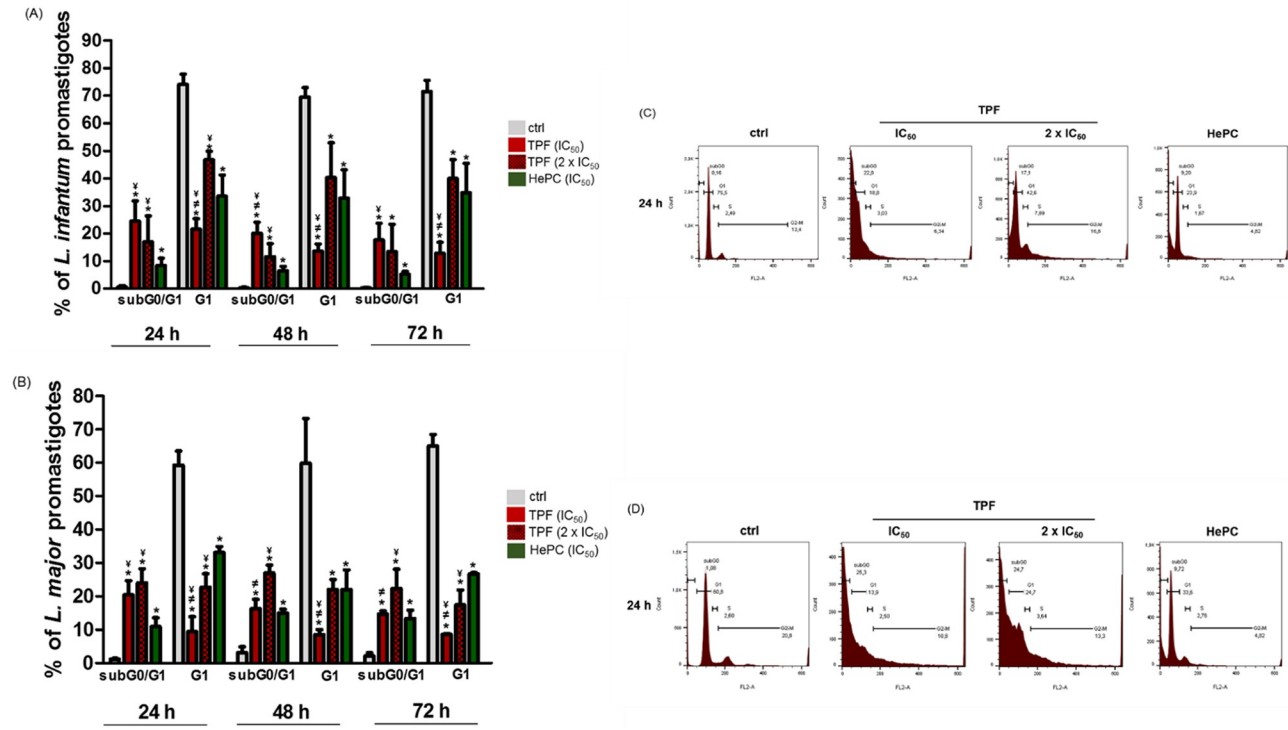

**Fig 5. Analysis of cell cycle arrest in TPF-treated *L. infantum* and *L. major* promastigotes.**

also significantly lower compared to untreated parasites for both *Leishmania* strains, while the respective proportions in the sub-G0/G1 region were higher (Fig 5A and 5B, and S2 Fig). Interestingly, for all the above time-points, proportions of TPF-treated promastigotes in the G1 region were equal or significantly lower compared to HePC-treated promastigotes, being in agreement with the observed respective high proportions in the sub-$G_0$ region.

Exponential-phase *L. infantum* (A, C) and *L. major* (B, D) promastigotes were either left untreated or were treated with $IC_{50}$ and 2 x $IC_{50}$ concentrations of TPF and HePC ($IC_{50}$) for 24, 48 and 72 h. Parasite cell cycle was analyzed by FACS and the results are presented as mean values ± SD in bar diagrams representative of three independent experiments (A, B). Data obtained after 24 h of treatment are also plotted as single parameter histograms representative of one experiment (C, D). Symbols of * and ¥ indicate statistically significant differences compared to negative and positive control groups, respectively, while ≠ indicates significant differences between TPF groups.

## TPF induces phosphatidylserine externalization in *L. infantum* and *L. major* promastigotes

Another biochemical feature of apoptosis is the translocation of phosphatidylserine (PS) from the inner to the outer leaflet of the plasma membrane [34]. In order to study whether *L. infantum* and *L. major* promastigote death triggered by TPF is via apoptosis or necrosis, TPF-treated promastigotes were double stained with FITC-conjugated annexin V, that binds to PS and the non-permeable PI that selectively enters necrotic cells. PS exposure and cell membrane permeability were quantified by FACS [34]. Annexin V and PI discriminate between early- [annexin V (+)–PI (-)] and late- apoptotic cells [annexin V (+)–PI (+)], as

well between necrotic [annexin V (-)–PI (+)] and live [(annexin V (-)–PI (-)] cells [39]. As it is shown in Fig 6, the proportions of both *L. infantum and L. major* promastigotes at early and late apoptotic phase, were significantly increased following TPF treatment, while the number of the respective viable parasites was dramatically decreased. More specifically, treatment of *L. infantum* promastigotes with TPF (IC$_{50}$, 2 x IC$_{50}$) for 24 h, induced a significant increase in the percentage of annexin V (+) cells that was 28.2% for IC$_{50}$ and 52% for 2 x IC$_{50}$ compared to 8.2% of untreated cells (Fig 6A). Moreover, at 48 and 72 h of treatment, a significant increase in the percentage of late apoptotic cells for both concentrations of TPF was observed indicating that when increasing drug exposure time, parasites undergo into the late phase of apoptosis (Fig 6A and S3 Fig). Accordingly, treatment of *L. major* promastigotes with TPF at all three time-points, also showed a significant increase in the percentages of annexin V (+) cells. At 24 h, untreated cells exhibited the percentage of 7.2% of annexin V (+) cells *vs* 50% for both IC$_{50}$ and 2 x IC$_{50}$ concentration (Fig 6B). Respectively, at 48 and 72 h of drug exposure, TPF-treated *L. major* promastigotes exhibited a significant increase in the percentage of annexin V (+) cells from 9.5% and 14% of untreated cells respectively, to 41% and 37% for the IC$_{50}$ concentration and 43.5% and 37% for the 2 x IC$_{50}$ concentration, respectively (Fig 6B). Representative flow cytometric dot plots with respective quadrants, representative of one experiment that illustrate early- and late- apoptotic cells, as well as necrotic cells are presented in Fig 6C and 6D, and S3 Fig.

It is also noteworthy that the effect of TPF on PS exposure was similar or even more potent than the relevant effect of HePC for both parasite strains at the three examined time-points. Nevertheless, TPF treatment caused also an increase in the percentages of necrotic cells compared to untreated cells, as well as to HePC-treated cells. The positive control of necrosis was consisted of promastigotes exposed to Triton X-100 and contained 72.24% and 46.9% of necrotic cells, for *L. infantum* and *L. major* respectively.

## TPF disrupts the mitochondrial membrane potential (*ΔΨm*) of *Leishmania* spp. promastigotes

The induced changes in the mitochondrial membrane potential of *L. infantum* and *L. major* promastigotes treated with TPF, were assessed using the mitochondrial membrane-permeable cationic potentiometric vital dye JC-1 which exhibits potential-dependent accumulation in mitochondria inversely related to *ΔΨm*. Net negative charge of mitochondrial membrane is characteristic of healthy cells thus allowing the concentration of the cationic dye. JC-1 aggregates emit red fluorescence (590 nm or FL-2) at higher potential whereas with membrane potentials below 140 mV remains as a monomer within the cytoplasm emitting green fluorescence (530 nm or FL-1) [26]. Consequently, mitochondrial depolarization is indicated by a decrease in the red/green fluorescence intensity ratio [40]. In order to assess changes in the *ΔΨm* upon exposure to TPF, we determined the ratio between red and green fluorescence (FL-2/FL-1 ratio) by FACS analysis. We showed that TPF provoked the loss of *ΔΨm* in both *L. infantum* and *L. major* promastigotes, as indicated by the significantly reduced FL-2/FL-1 ratio, in our experimental conditions (Fig 7). TPF caused depolarization of *ΔΨm* within 24 h of treatment in both *L. infantum* and *L. major* promastigotes. The FL-2/FL-1 ratio was significantly affected, in *L. infantum* promastigotes in a dose-independent manner. Indicatively, TPF at both concentrations induced an almost 85% reduction of the ratio at 24 h of treatment (Fig 7A). The effect of TPF on the FL-2/FL-1 ratio was also time-independent since these alterations remained at the same levels after 48 and 72 h of treatment. Similarly, the FL-2/FL-1 ratio was significantly affected, in *L. major* promastigotes also in a dose- and time-independent manner. Indicatively, TPF at the IC$_{50}$ concentration induced an 83%

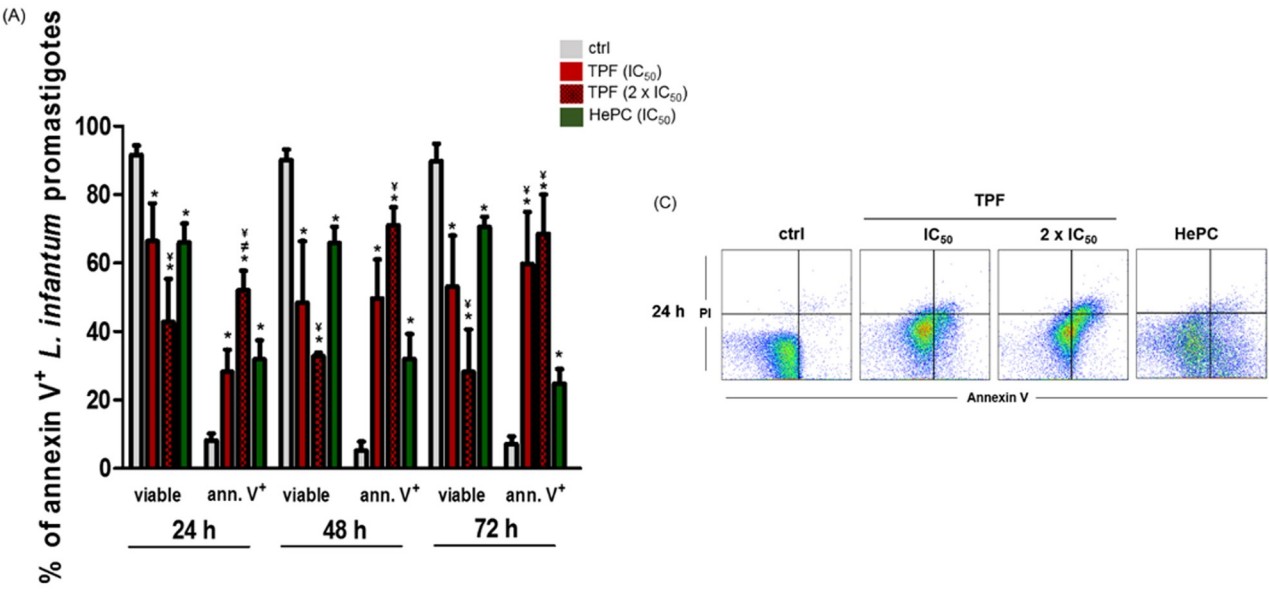

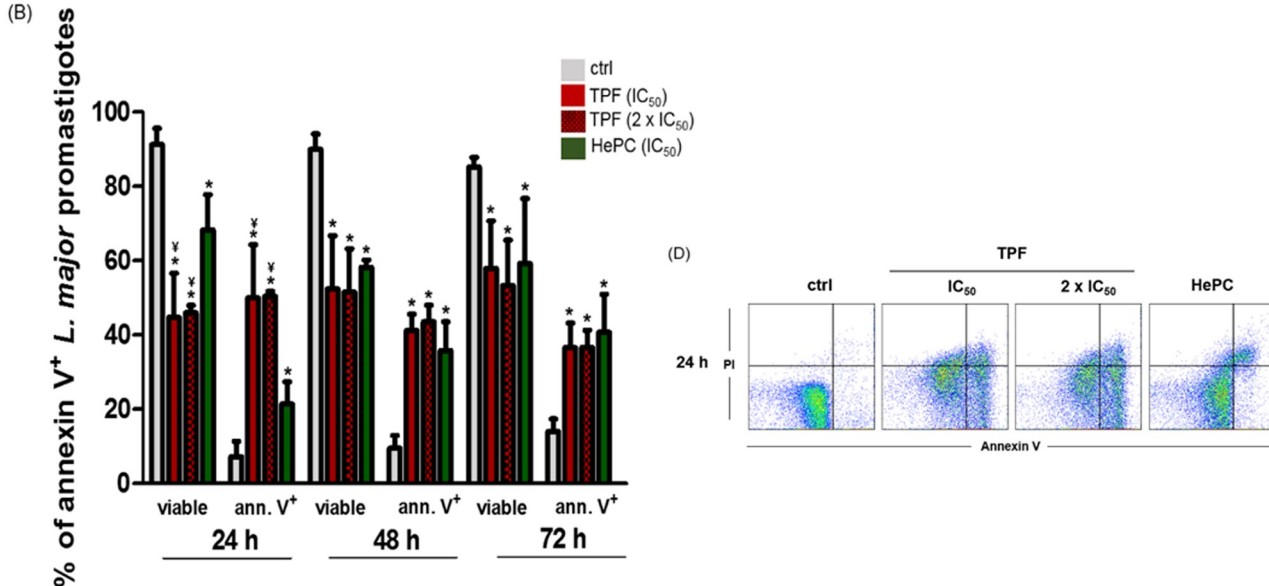

**Fig 6. PS externalization in TPF-treated *L. infantum* and *L. major* promastigotes.** Exponential-phase *L. infantum* (A, C) and *L. major* (B, D) promastigotes were either left untreated (negative control) or were treated with $IC_{50}$ and $2 \times IC_{50}$ concentrations of TPF and HePC ($IC_{50}$, positive control) for 24, 48 and 72 h. At the end of the aforementioned time-points, parasites were double stained with annexin V-FITC and PI and were analyzed by FACS. The results are presented as mean values of % annexin V+ parasites ± SD in bar diagrams (A, B) representative of three independent experiments and as flow cytometric dot plots with respective quadrants (C, D), representative of one experiment. Symbols of * and ¥ indicate statistically significant differences compared to negative and positive control groups, respectively while ≠ indicates significant differences between TPF groups.

reduction of the ratio and this value was 76% with the $2 \times IC_{50}$ concentration, at 24 h of treatment (Fig 7B). It is noteworthy that the effect of TPF on the FL-2/FL-1 ratio was similar with the respective effect of the reference drug HePC for both *Leishmania* strains. Furthermore, the depolarization of the mitochondria evidenced by the $\Delta\Psi m$ fall was supported by the shift

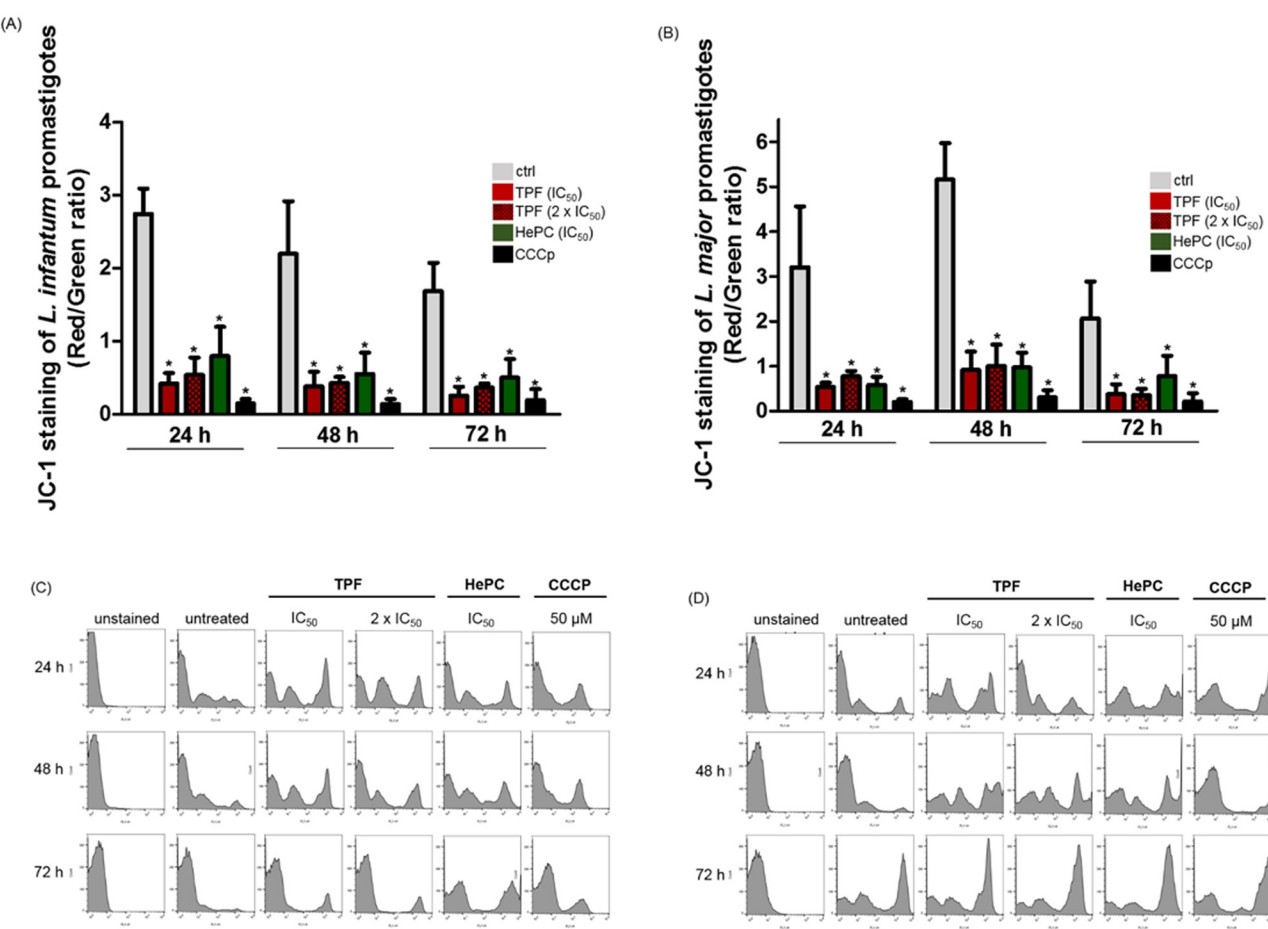

**Fig 7. Evaluation of mitochondrial membrane potential in *Leishmania* spp. promastigotes treated with TPF.**

of both *Leishmania* populations towards the right in the FL-1 channel (green fluorescence) as observed in the representative FACS histograms (Fig 7C and 7D).

Exponential-phase *L. infantum* (A, C) and *L. major* (B, D) promastigotes were treated with $IC_{50}$ and 2 x $IC_{50}$ concentrations of TPF and HePC ($IC_{50}$) for 24, 48 and 72 h. JC-1 dye was added at a final concentration of 2 μM and parasites were incubated in the dark for 20 min and washed to eliminate the non-internalized dye. Fluorescence intensity was determined by FACS (FL-2: J-aggregates and FL-1: JC-1 monomers). Untreated promastigotes and promastigotes treated with CCCP (50 μM), were used to achieve minimal and maximal depolarization, respectively. Results are presented as mean values ± SD of FL-2/FL-1 ratio in bar diagrams (A, B) representative of three independent experiments and as single parameter histograms (FL1-H) (C, D) representative of one experiment. Symbol of * indicates statistically significant differences compared to untreated parasites.

## TPF provokes the generation of intracellular reactive oxygen species (ROS) in *Leishmania* spp. promastigotes

ROS play a central role in cell signaling, as well as in regulation of the main pathways of apoptosis mediated by mitochondria, death receptors and the endoplasmic reticulum. Excess cellular levels of ROS cause damage to proteins, nucleic acids, lipids, membranes and organelles such as

mitochondria, which can lead to activation of cell death processes such as apoptosis [41]. In order to estimate whether TPF treatment triggered *in vitro* the intracellular levels of ROS in *Leishmania* spp. promastigotes, the cell permeant fluorescent probe $H_2DCFDA$ was used and analyzed by flow cytometry. TPF triggered an increase of cytosolic ROS production in a dose- and time- dependent manner in both *L. infantum* and *L. major* promastigotes (Fig 8). Our data stated about a 10-fold and 20-fold increase in fluorescence intensity of *L. infantum* promastigotes treated with the $IC_{50}$ and 2 x $IC_{50}$ concentration of TPF, respectively, compared to untreated parasites, after a 24h exposure (Fig 8A and 8B). The respective increase at 48 and 72 h reached the 15- and 30-fold for the $IC_{50}$ and 2 x $IC_{50}$ concentration and 30- and 50-fold, respectively (Fig 8A and 8B). Accordingly, the effect of TPF treatment in cytosolic ROS production in *L. major* promastigotes was also time-dependent. Analytically, ROS concentration in *L. major* promastigotes reached an 8-fold and 9-fold production in the presence of the $IC_{50}$ and 2 x $IC_{50}$ concentration of TPF respectively, compared to untreated parasites, after 24 h exposition (Fig 8C and 8D), while the relevant increase at 48 h reached a 15- and 19-fold for $IC_{50}$ and 2 x $IC_{50}$

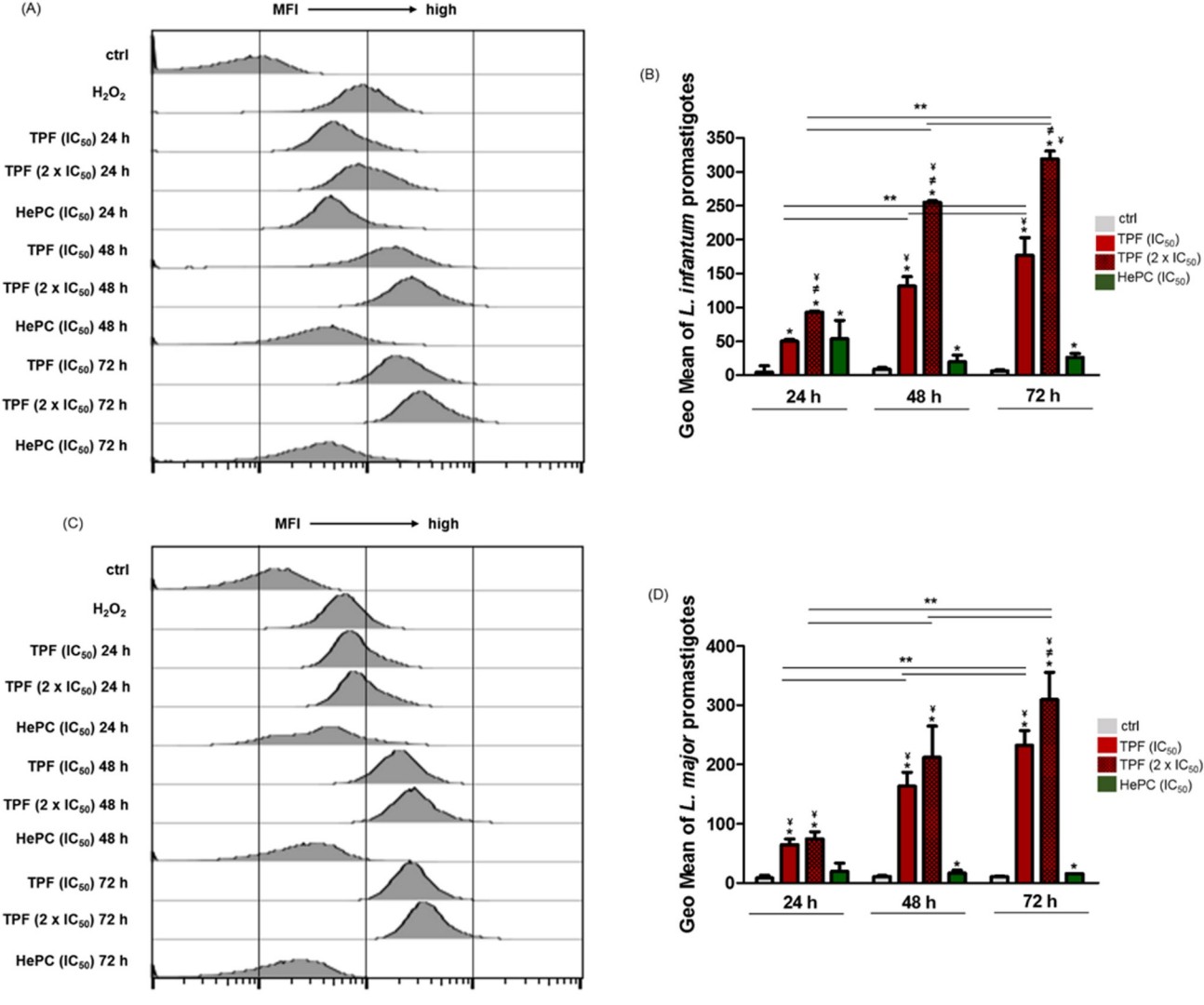

**Fig 8. ROS production in TPF-treated *Leishmania* spp. promastigotes.**

concentration and 22- and 30-fold, respectively at 72 h (Fig 8C and 8D). Positive control of ROS generation was consisted of *L. infantum* and *L. major* promastigotes exposed to $H_2O_2$.

Exponential-phase *L. infantum* (A, B) and *L. major* (C, D) promastigotes were either left untreated or were treated with $IC_{50}$ and 2 x $IC_{50}$ concentrations of TPF and HePC ($IC_{50}$). The cellular ROS levels were quantified at 24, 48 and 72 h using $H_2$DCFDA and flow cytometry. Data are representative of three independent experiments and results are presented as single parameter histograms representative of one experiment (A, C) and as fluorescence intensity mean values ± SD in bar diagrams (B, D). Symbols of * and ¥ indicate statistically significant differences compared to untreated and HePC-treated parasites respectively, while ≠ indicates significant differences between TPF groups and ** indicate differences among time-points.

## TPF effectively reduces parasite load in popliteal lymph nodes

In our previous study, we demonstrated the therapeutic effect of TPF in experimental cutaneous leishmaniasis as it was indicated by a significant reduction of footpad swelling during the infection course [14]. In the present study, in order to unravel further the therapeutic effect of TPF on cutaneous leishmaniasis and to quantitatively correlate its effect with the number of viable parasites in popliteal lymph node cells, BALB/c mice were challenged with *L. major* promastigotes and followed the treatment schedule as it is illustrated in Fig 9A. Mice were weighted once a week from the beginning until the end of the experiment and we ascertained no fluctuation, indicating their well-being. Moreover, the infection progress was monitored by measuring the increase in footpad swelling weekly (Fig 9B). The obtained data clearly revealed that TPF treatment induced significant therapeutic effect reaching an 84% (p = 0.003) reduction of parasite load in popliteal lymph nodes compared to *L. major*—infected and untreated mice (7433 ± 149 *vs* 43918 ± 7665 parasites/organ) (Fig 9C) at five weeks post-treatment termination. HePC also exhibited significant reduction of parasite load reaching 60% (p = 0.020) compared to untreated mice (17460 ± 6880 *vs* 43918 ± 7665 parasites/organ) (Fig 9C). At last, it is important to state that the therapeutic effect of TPF was more prominent than the relative effect of HePC (p = 0.020).

## TPF treatment results in lower *Leishmania*-specific IgG1 antibody production

The antibody isotype profile is correlated with the protective and non-protective against leishmaniasis, Th1 and Th2 CD4+ T cell differentiation, respectively. Switching of the IgG isotype to IgG2a is provoked by IFN-γ, a Th1 cytokine, while the IL-4, a Th2 cytokine, is correlated with the IgG1 production [42,43]. Assessment of *Leishmania*-specific humoral responses five weeks post-treatment termination, showed that *L. major* infected-mice treated with TPF developed significantly lower (p = 0.042) titers of IgG1-specific antibodies compared with infected and untreated control mice, while IgG2a-specific antibodies were observed at equal levels (Fig 9D). This antibody isotype pattern reflects the predominance of a Th1-type immune response.

## TPF enhances the IFN-γ production by CD4+ T lymphocytes

Th1-type host immune responses are pivotal for ascertaining the successful outcome of the tested treatment against murine cutaneous leishmaniasis. IFN-γ secretion is associated with Th1-type immunity while IL-4 is a Th-2-polarizing stimulus [44]. Moreover, IFN-γ has a critical role in the activation of macrophages to kill intracellular pathogens such as Leishmania spp. In an attempt to better understand the immunomodulatory effects of TPF, we performed flow cytometric analysis for the detection of parasite-specific IFN-γ- and IL-4-producing CD4 + T cells in stimulated spleen cell cultures at the respective time-point. Mice treated with TPF

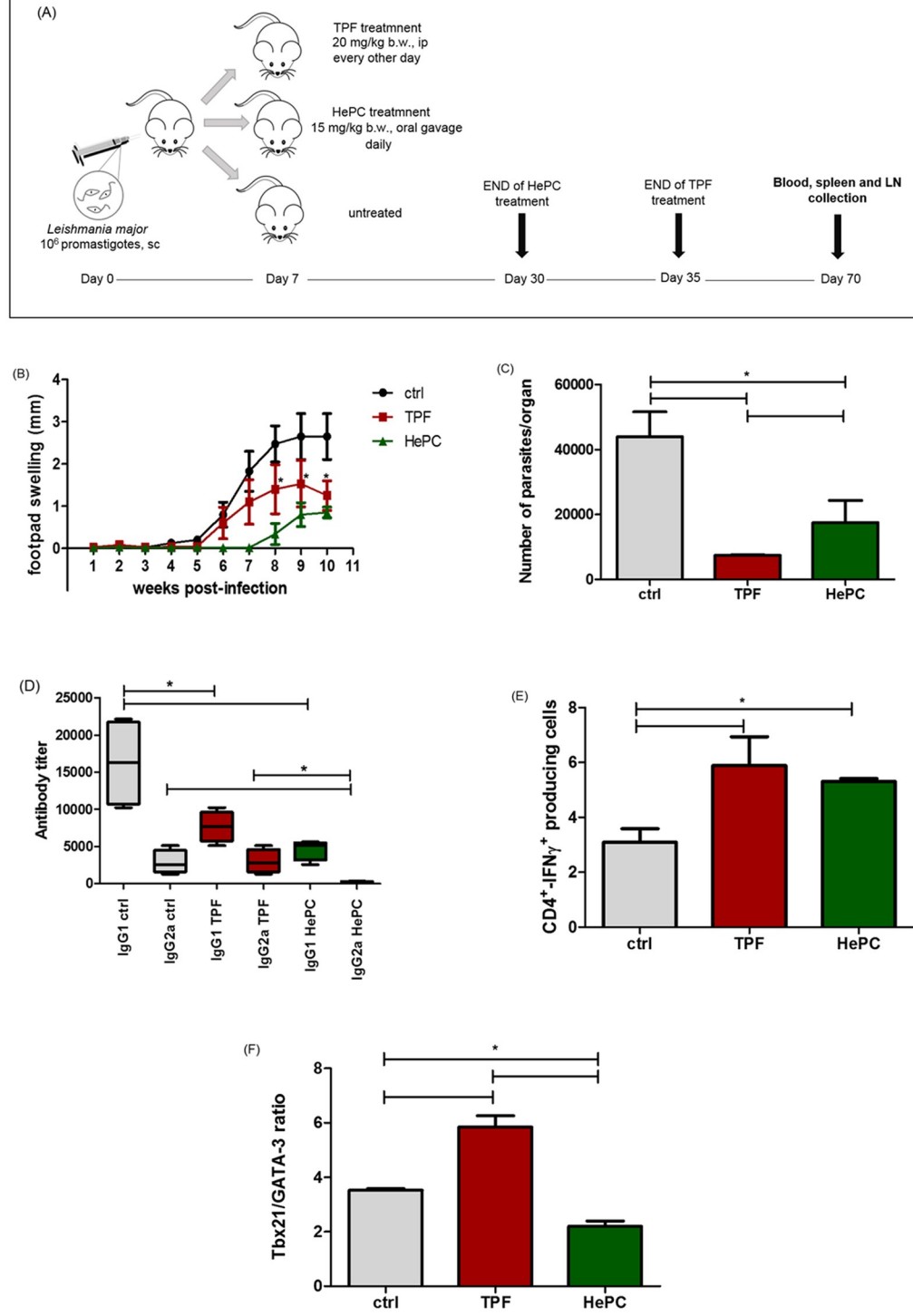

**Fig 9. Therapeutic effect of TPF in a murine experimental model of cutaneous leishmaniasis. (A) Definition of treatment schedule and experimental groups**. BALB/c mice were subcutaneously infected into the right hind footpad with $10^6$ *L. major* promastigotes and one week later, were either treated with TPF (20 mg/kg b.w., ip) or HePC (15 mg/kg b.w., oral gavage) for up to 28 days or left untreated (control group). **(B) Time course of *L. major* infection**. The increase in footpad swelling was monitored weekly by measuring the increase of footpad thickness (in mm) as compared with the uninfected contralateral footpad. Results are expressed as mean values ± SD. **(C) Parasite load estimation in popliteal lymph nodes of TPF-treated mice**. Parasite load was determined at five weeks post-treatment termination by a limiting dilution assay. Values represent the mean value ± SD of five animals per experimental group. **(D) The IgG1**

**and IgG2a reciprocal end-point titers** against *L. major* SLA were determined by ELISA at five weeks post-treatment termination (n = 5 mice per group) and represented as whisker (min to max) plots. * indicates the statistical differences between IgG1 and IgG2a anti-SLA titers among experimental groups. **(E) Effect of TPF treatment on the percentage of IFN-γ-producing CD4+ T cells**. Spleen cells from each experimental animal were isolated five weeks post-treatment termination and intracellular staining was performed. Flow cytometric analysis was conducted and total splenocytes were first gated on a forward scatter (FSC-H)/side scatter (SSC-H) plot and then subgated on the CD4+ population. Results of cytokine profiles are presented in bar diagram showing the percentage of IFN-γ-producing CD4+ T cells. Symbol of * indicates statistically significant differences among the experimental groups. **(F) Quantitative analysis of Tbx21 and GATA-3 mRNA in spleen cells**. The expression of Tbx21 and GATA-3 mRNA was evaluated by real-time PCR and the expression level of the GAPDH housekeeping gene from the same samples was used as normalizer. All expression levels were computed via the *ΔΔCt* method. Symbol of * indicates statistically significant differences among the experimental groups.

exhibited significant elevated levels of CD4+ T cells able to produce IFN-γ, upon in vitro re-stimulation with SLA, compared to infected and untreated mice (5.89 ± 1.8% vs 3.1 ± 0.8%, p = 0.028) (Fig 9E). It is worth mentioning that mice treated either with TPF or HePC exhibited similar levels of IFN-γ-producing CD4+ T cells (p = 0.686) (Fig 9E). The frequency of IL-4-producing CD4+ T cells in TPF-treated mice was similar to infected and untreated mice.

## Tbx21/GATA-3 gene expression ratio is upregulated in TPF-treated mice

Among regulatory signals that crucially influence the CD4+ T cell differentiation, the transcription factors Tbx21 and GATA-3 are suggested as master regulators of Th1 and Th2 polarization, respectively [45]. Thus, Tbx21/GATA-3 ratio provides a useful tool to evaluate the immune balance between the Th1 and Th2-type responses [46]. To this end, spleen tissue was collected at five weeks post-treatment termination and the expression of mRNAs encoding for Tbx21 (T-bet) and GATA-3 transcription factors was analyzed. Our data showed that spleen tissue samples obtained from TPF-treated mice exhibited higher Tbx21/GATA-3 gene expression ratio (Fig 9F) compared to infected and untreated mice (+ 5.9 fold vs + 3.7 fold, respectively, p = 0.050), indicating a shift toward the Th1 differentiation.

## Discussion

During the last decade, the interest in the quest for natural products which concomitantly exert chemotherapeutic and immunomodulatory properties, has been reinforced, due to the occurrence of numerous adverse effects of the existing drugs [47]. We have previously demonstrated the *in vitro* anti-leishmanial properties of TPF derived from EVOO, against both *Leishmania* developmental stages, the cell-free promastigotes and the intracellular amastigotes, as well as its ability to modulate the host's immune system by mediating the increased production of ROS and NO in *Leishmania*-infected J774A.1 macrophages [14]. We have also validated its therapeutic potency in an *in vivo* murine experimental model of cutaneous leishmaniasis since TPF-treated mice exhibited significant reduction of footpad swelling [14]. These results prompted us to further establish its anti-promastigote activity against both viscerotropic and dermotropic *Leishmania* strains and to delineate the relevant cell death mechanism by which TPF induces its anti-leishmanial potency. To our knowledge no similar study on TPF, an essential component of olive oil, has been reported before.

The anti-leishmanial activity of TPF was tested against *L. infantum* and *L. major* promastigotes. Their viability after treatment with various increasing concentrations of TPF was evaluated using the resazurin reduction assay based on the ability of resazurin (oxidized blue form) to be irreversibly reduced by enzymes in viable cells and to generate a red fluorescent resorufin product while a reduction in the number of metabolically active cells indicates a decrease in cell viability. Next, the anti-leishmanial effect of TPF on *L. infantum* and *L. major* parasites

was further validated in early exponential-phase promastigote cultures by cell counting and flow cytometric approaches, over a time course of up to 72 h. The rapid and precise determination of proliferation rates and generation times by CFSE staining is beneficial for the analysis of the effects of anti-leishmanial compounds [48]. TPF proved to be effective on both viability and proliferation of *L. infantum* and *L. major* promastigotes. Obtained results indicated a clear inhibition of promastigote multiplication rate and a decelerated fluorescence decrease in TPF-treated, compared to untreated parasites. An almost stable fluorescence intensity during the days of treatment, suggested an almost completely arrested cell proliferation. Similar studies have reported the use of these approaches for the determination of the multiplication rate of *Leishmania* parasites after exposure to amiodarone, an antiarrhythmic drug used for the symptomatic treatment of chronic Chagas' disease patients with cardiac compromise, and allopurinol, a purine analogue used to decrease high blood uric acid levels but also to treat canine leishmaniasis [49,50]. We also evaluated alterations in cell shape and size of promastigote forms, after treatment with TPF. The round or ovoid shaped parasites exhibiting higher fluorescence intensity were assessed by confocal fluorescence microscopy. The observed morphological changes were similar to alterations induced by other natural products. *L. donovani* promastigotes shrank and became aflagellated and oval or round, after exposure to racemoside A, a natural steroidal saponin [51] and HePC [38]. Also, it has been reported that yangambin, a lignan obtained from *Ocotea duckei*, altered the morphology of *L. chagasi* and *L. amazonensis* promastigotes in the same way [52], while another study for the trigger of apoptosis by essential oils from plants of the genus *Artemisia* reported shrinkage of *L. infantum* promastigote bodies [53].

Moreover, in the present study, we offer detailed insights into the mechanism of action of TPF by evaluating basic biochemical features associated with apoptosis. Analysis of DNA content reveals cell ploidy and provides information on cell position in the cell cycle, allowing the estimation of frequency of apoptotic cells that are characterized by fractional DNA content [54]. Apoptotic cells generally feature active endonucleases that preferentially cleave DNA, translating into an increased cell population located proximal to the G0-G1 peak, often being defined as "sub-G0/G1" cell population [55]. We demonstrated here for the first time to our best knowledge, that TPF is able to induce sub-G0/G1 cell cycle arrest of *Leishmania* spp. promastigotes. Flow cytometric analysis revealed that both exponential-phase *L. infantum* and *L. major* promastigotes enter into the sub-G0/G1 stage and get arrested therein, as indicated by the increased percentages of TPF-treated promastigotes in the sub-G0/G1 region and the following decrease in G1 region. This phenomenon was observed even at 24 h of exposure, in both *Leishmania* species tested. The TPF treatment proved to be more potent to halt the cell cycle in the sub-G0/G1 phase compared to the effect of HePC. The oral drug HePC has been previously published to cause sub-G0/G1 cell cycle arrest in *L. donovani* promastigotes [38] and the fact that TPF acts more effectively to the same direction, may be promising. Similarly, the cell cycle arrest in the sub-G0/G1 phase has been also observed, in other studies with natural products. The established anti-malarial artemisinin, a sesquiterpene lactone isolated from *Artemisia annua*, as well as, extracts of *Artemisia annua* leaves and seeds, have been reported to cause cell cycle arrest in *L. donovani* parasites [37,56]. Similarly, plant extracts obtained from *Allium sativum* and *Valeriana wallichii* also induced enhanced proportion of *Leishmania* spp. parasites in the sub-G0/G1 phase [57,58]. Additionally, Kalsome TM10, a liposomal amphotericin B, induced a dose- and time-dependent arrest of *L. donovani* promastigotes, in the sub-G0/G1 phase [59].

Two main plasma membrane alterations have been described during cell death processes, the PS externalization and the permeabilization to PI. PS is predominantly located in the inner leaflet of the plasma membrane, whereas, upon initiation of apoptosis, it is translocated to the

extracellular membrane leaflet where it identifies cells as targets for phagocytosis [60,61]. At the early event of apoptosis, the plasma membrane still excludes viability assays such as PI, so single staining with annexin V serves as an early marker of apoptosis, while in the late stage of apoptosis, the membrane loses its integrity and the late apoptotic/secondary apoptotic cells, become positive for both annexin V and PI [55,62]. Consequently, the absence of PI staining signals membrane integrity while necrotic cells become PI-permeable even though they remain negative to annexin V. However, it should be acknowledged that the exposure of PS in *Leishmania* promastigotes as a marker of apoptosis-like cell death is still a controversial subject [63]. Research models based on the detection of PS exposure suggest that *Leishmania* parasites mimic mammalian apoptotic cells by exposing PS at their cell surface to trigger phagocytic uptake into host macrophages [64–67]. Nevertheless, annexin V is not specific for PS and also binds other phospholipids such as phosphatidylglycerol (PG) and phosphatidylinositol-4,5-bisphophate [68]. Previous studies on *Leishmania* lipid compositions by chromatography-based methods have reported the presence of PS in several *Leishmania* species [69–71], while other studies based on mass spectrometry analysis failed to detect this lipid [72–75]. Taking into consideration that some authors affirm that primary necrotic cells with intensely damaged membranes are stained rapidly and strongly with PI but not with annexin V, our results suggest that TPF induced *Leishmania* spp. cell death by apoptosis, as was indicated by high exposure of annexin V binding phospholipids on the surface of the parasites. Treatment of *L. infantum* promastigotes with TPF, exhibited a gradual increase in percentages of late apoptotic cells, indicating that when increasing drug exposure time, parasites undergo into the late phase of apoptosis. Accordingly, treatment of *L. major* promastigotes with TPF at all three time-points, also showed a significant increase in the percentages of annexin V positive cells. Previous studies have highlighted PS exposure after treatment with several anti-leishmanial drugs. *Artemisia annua* leaf essential oil enhanced mainly the proportions of single stained with annexin V late-log phase *L. donovani* promastigotes [76], while the apoptosis-like death induced in *L. donovani* promastigotes by EROSA, an eugenol-rich oil derived from *Syzygium aromaticum*, was characterized by both annexin V and PI binding [55].

Mitochondria take part in a number of fundamental cellular processes, including energy production, biosynthetic pathways and cellular oxidoreductive homeostasis while their dysfunction can lead to numerous pathophysiological consequences [77]. It has been demonstrated that nuclear features of apoptosis in unicellular protozoan cells, as in metazoan cells, is characterized by condensation of nuclei and fragmentation of DNA, that are preceded by alterations in mitochondrial structure and transmembrane potential [27]. *Leishmania* spp. possess a single mitochondrion that responds to energy requirements and consequently it represents a potential drug target [78]. The loss of mitochondrial membrane potential has been observed to play key role in drug-induced death in protozoans such as *Leishmania* spp. [79]. Thus, we tested the effect of TPF on possible alterations in mitochondrial transmembrane potential of *Leishmania* spp. promastigotes and our data provided evidence that TPF induced a remarkable decrease in mitochondrial membrane potential within 24 h of treatment, as was reflected by a drastic fall in the red/green fluorescence intensity ratio.

Thus, while mitochondrial depolarization is critical, it is the subsequent elevation in the levels of oxidizing species generated from the mitochondria that plays the role of cytotoxic effectors in apoptosis [80,81]. Mitochondrial ROS is the very important potential ROS pool, sealed in the double membranes of mitochondria. In *Leishmania* parasites, there is a basal level of ROS maintained by the mitochondria inside the cells for physiological signaling, while the onset of mitochondrial dysfunction causes leakage in the electron transport chain and thus elevates ROS levels [82,83]. We found that TPF treatment promoted cytosolic ROS generation in a dose- and time- dependent manner in both *L. infantum* and *L. major* promastigotes. Several

similar studies suggest an increase in cellular ROS production, being responsible for the apoptotic process in *Leishmania* spp., triggered by various natural products. For example, LQB-118, an orally active pterocarpanquinone and an eugenol-rich oil of *Syzygium aromaticum* triggered the induction of excessive ROS production in *L. amazonensis* and *L. donovani* promastigotes, respectively [55,84].

Taken together, reduction of proliferation rate, morphological alterations, cell population arrest at sub-G0/G1 region, PS exposure, alterations in mitochondrial potential and excessive ROS production, all hallmarks of classic apoptosis, indicate an apoptotic-like cell death in *L. infantum* and *L. major* promastigotes caused by TPF.

The *in vitro* obtained data prompted us to evaluate the therapeutic effect of TPF in an experimental model of cutaneous leishmaniasis in BALB/c mice. As mentioned above, we had previously demonstrated the therapeutic effect of TPF against *L. major*-infected BALB/c mice as indicated by a significant reduction of footpad swelling during the course of infection [14]. The observed lesions in TPF-treated mice were diminished compared to the control group of infected and untreated mice which reached profound skin lesions and even necrosis. In the present study, this therapeutic effect was further verified and linked to significant reduction of parasite load in the draining lymph nodes at the site of infection. Indeed, the therapeutic treatment with TPF that considerably minimized the parasite burden was based on the half number of injections compared to the reference drug, since TPF was administered every two days, whereas HePC was administered daily. So far, the majority of the chemotherapeutic treatments against leishmaniasis used for decades are long lasting, either by oral administration, as HePC or by parenteral administration, as meglumine antimoniate [2]. The weight of all TPF-treated mice remained stable during the experimental process indicating indirectly both the non-toxicity of TPF, as well as the sustained homeostasis over the controlled parasitic infection. Our results are in accordance with a plethora of other studies which report effective decrease in parasite load upon treatment with natural products either in cutaneous [85–87] or in visceral leishmaniasis experimental mouse models [13,88–90].

Leishmaniasis is an immune-mediated disease and thus the concomitant potentiation of Th1 cell activation through the use of immunomodulators in addition to conventional chemotherapy has been hypothesized as a future therapeutic option for the disease [91]. The murine model of *L. major* infection was originally used to define the Th1/Th2 T-cell polarization and revealed the importance of the Th1/Th2 balance in favor to the Th1-type response for the successful control over the infection. In order to achieve disease resolution, favoritism over the Th1 immune response must be selectively triggered by therapeutic schemes and/or compounds capable to serve as immunomodulators [92]. Therefore, focusing over the immunomodulatory properties of TPF, the critical issue was to assess its potential to modulate the immune response of *L. major*-infected BALB/c mice towards the establishment of Th1-type immune response capable to restrain the parasite expansion. Firstly, the *in vivo* immunomodulatory effects of TPF were probed in terms of decreased levels of Th2-type immune response as depicted in lower *Leishmania*-specific IgG1 antibody titer in TPF-treated mice compared to infected and untreated controls. Given the observed relationship between the levels of IgG1 and IgG2a antibody isotypes with a bias towards the non-protective Th2 and the healing Th1 responses, respectively [93,94], the highest IgG2a/IgG1 ratio observed in TPF-treated mice, suggests that TPF is able to direct the driven by *Leishmania* infection, Th2-type immune response to the protective Th1-type. Moreover, TPF affects markers of the host Th1-type response, as evidenced by the enhanced IFN-γ producing CD4[+] T cell population at five weeks post-treatment termination. The activation of such effector cells that produce the IFN-γ macrophage-activating cytokine is necessary for the effective host control over parasite replication. A similar study demonstrated that treatment of *L. donovani*-infected BALB/c mice with *Croton*

*caudatus* leaf extract also induced enhanced population of IFN-γ producing CD4$^+$ T cells in splenocytes of treated mice [95]. Furthermore, the immunomodulatory effect of TPF treatment on the Th1/Th2 differentiation was further appraised in the transcription levels of Tbx21 and GATA-3 transcription factors. Tbx21/GATA-3 ratio has been demonstrated as a surrogate marker of Th1/Th2 cytokine balance [46] and our data revealed that TPF treatment significantly enhanced this transcript ratio in spleen cells compared with infected and untreated controls, also reflecting the predominance of a Th1-type immune response. Several other extracts from different plants, display a wide variety of pharmacological activities, which may not only be due to their direct leishmanicidal action on the parasite, but also to their concomitant effect on enhancing the host's protective immune response. Similar studies in *in vitro* and *in vivo* experimental models of visceral leishmaniasis have reported the ability of *Xylopia discreta* extract [96], and ethyl acetate extract fraction from *Azadirachta indica* [97] to influence the host immunity through the stimulation of Th1-type immune response.

Overall, these findings establish that TPF exhibits anti-leishmanial activity which is not only due to its direct action on the parasite but also to a simultaneous effect on the host immune response by inducing distinct T cell subpopulations that affect immunoregulation in favor to host protection. Additionally, our results contribute to the research for the promising impact of EVOO on human health while also validate our previous findings for the potential use of olive tree natural products as candidate drugs against leishmaniasis in terms of being affordable and safe.

## Supporting information

**S1 Fig. *Leishmania* spp. promastigote proliferation determined by CFSE staining.** *L. infantum* and *L. major* early exponential-phase promastigotes were treated with IC$_{50}$ and 2 x IC$_{50}$ concentrations of TPF and their proliferation rate was qualitatively monitored at 24 h intervals for 3 consecutive days by CFSE staining and subsequent analysis of fluorescence intensity in FACS. HePC (IC$_{50}$)-treated and untreated parasites were used as positive and negative control groups, respectively. The results are presented as single parameter histogram overlays representative of one experiment.
(TIF)

**S2 Fig. Analysis of cell cycle arrest in TPF-treated *L. infantum* and *L. major* promastigotes.** Exponential-phase *L. infantum* and *L. major* promastigotes were either left untreated or were treated with IC$_{50}$ and 2 x IC$_{50}$ concentrations of TPF and HePC (IC$_{50}$) for 48 and 72 h. Parasite cell cycle was analyzed through FACS and the results are plotted as single parameter histograms representative of one experiment.
(TIF)

**S3 Fig. PS externalization in TPF-treated *L. infantum* and *L. major* promastigotes.** Exponential-phase *L. infantum* and *L. major* promastigotes were either left untreated (negative control) or were treated with IC$_{50}$ and 2 x IC$_{50}$ concentrations of TPF and HePC (IC$_{50}$, positive control) for 48 and 72 h. At the end of the aforementioned time-points, parasites were double stained with annexin V-FITC and PI and were analyzed by FACS. The results are presented as flow cytometric dot plots with respective quadrants, representative of one experiment.
(TIF)

## Acknowledgments

We thank Dr Evangelia Xingi and the members of the Light Microscopy Unit of the Hellenic Pasteur Institute for their help with the microscopy work.

## Author Contributions

**Conceptualization:** Kalliopi Karampetsou, Olga S. Koutsoni, Leandros-Alexios Skaltsounis, Eleni Dotsika.

**Formal analysis:** Kalliopi Karampetsou, Olga S. Koutsoni.

**Funding acquisition:** Kalliopi Karampetsou, Eleni Dotsika.

**Investigation:** Kalliopi Karampetsou, Olga S. Koutsoni, Georgia Gogou.

**Methodology:** Kalliopi Karampetsou, Olga S. Koutsoni, Eleni Dotsika.

**Project administration:** Eleni Dotsika.

**Resources:** Apostolis Angelis, Leandros-Alexios Skaltsounis.

**Supervision:** Eleni Dotsika.

**Validation:** Eleni Dotsika.

**Visualization:** Kalliopi Karampetsou, Olga S. Koutsoni, Apostolis Angelis, Leandros-Alexios Skaltsounis, Eleni Dotsika.

**Writing – original draft:** Kalliopi Karampetsou, Olga S. Koutsoni, Eleni Dotsika.

**Writing – review & editing:** Kalliopi Karampetsou, Olga S. Koutsoni, Apostolis Angelis, Leandros-Alexios Skaltsounis, Eleni Dotsika.

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
