## [Decision Letter · Decision Letter 0]

27 Aug 2020

Dear Dr DOTSIKA,

Thank you very much for submitting your manuscript "Total Phenolic Fraction (TPF) from Extra Virgin Olive Oil: induction of apoptotic-like cell death in Leishmania spp. promastigotes and in vivo potential of therapeutic immunomodulation." for consideration at PLOS Neglected Tropical Diseases. As with all papers reviewed by the journal, your manuscript was reviewed by members of the editorial board and by several independent reviewers. In light of the reviews (below this email), we would like to invite the resubmission of a significantly-revised version that takes into account the reviewers' comments. 

The authors need to revise the manuscript as per the two reviewer's comments . The authors need to recalculate the data and also perform the experiments suggested by the reviewers. In addition, the authors need to rewrite the manuscript so that it is legible.

We cannot make any decision about publication until we have seen the revised manuscript and your response to the reviewers' comments. Your revised manuscript is also likely to be sent to reviewers for further evaluation.

Sincerely,

Hira L Nakhasi

Associate Editor

S Madison-Antenucci

Deputy Editor

The authors need to revise the manuscript as per the two reviewer's comments . The authors need to recalculate the data and also perform the experiments suggested by the reviewers. In addition, the authors need to rewrite the manuscript so that it is legible.

Reviewer's Responses to Questions

**Key Review Criteria Required for Acceptance?**

**Methods**

-Are the objectives of the study clearly articulated with a clear testable hypothesis stated?

-Is the study design appropriate to address the stated objectives?

-Is the population clearly described and appropriate for the hypothesis being tested?

-Is the sample size sufficient to ensure adequate power to address the hypothesis being tested?

-Were correct statistical analysis used to support conclusions?

-Are there concerns about ethical or regulatory requirements being met?

Reviewer #1: (No Response)

Reviewer #2: The objectives of the present work ar clear and the technical approaches used to test the hypothesis are appropriate to address it. A first objective is to confirm that total phenolic fraction TPF extrated from virgin olive oil has a leishmanicidal effect both in vivo and in vitro. Further, a second objective is to study the leishmanicidal mechanism of TPF, that is associated with the induction of apoptosis of the parasite and also with some immunomodulatory capability of TPF observed in the experimental model of infection.

**Results**

-Does the analysis presented match the analysis plan?

-Are the results clearly and completely presented?

-Are the figures (Tables, Images) of sufficient quality for clarity?

Reviewer #1: (No Response)

Reviewer #2: The analysis one are in agreement with the objectives proposed, the results are presented clearly and the figures included in the manuscript are of sufficient quality for clarity. Indeed, some of the FACS images included in the figures can be eliminated for the article publication, keeping just the bar graphics. Such FACS images can be included in a supplemmentary data file.

**Conclusions**

-Are the conclusions supported by the data presented?

-Are the limitations of analysis clearly described?

-Do the authors discuss how these data can be helpful to advance our understanding of the topic under study?

-Is public health relevance addressed?

Reviewer #1: (No Response)

Reviewer #2: The results obtained support the conclusions in a consistent way. Some limitations of the study are described, mainly in the in vivo model. The importance of the results are highlighted by the authrs and they are relevant in the context of public health because of the need to develop and evaluate new antileishmanial drugs.

**Editorial and Data Presentation Modifications?**

Reviewer #1: (No Response)

Reviewer #2: Minor revisions

-As mentioned before, FACS images included in the figures together with the barr graphic are redundant. Such figures can be included in the supplementary data file.

- Regarding experimental infection in mice, data on the swelling of the footpad from infected mice should be included in the manuscript. I know similar results were obtained in previous works already published, but it is interesting to show how the swelling of footpad correlated with those data of parasite burden observed in the lymph node. Further, the authors mention that swelling of the footpad is different in control and TPF treated infected mice, being less severe and without necrosis, a new figure with pictures of the footpad showing such differences in the lesions might be included.

**Summary and General Comments**

Reviewer #1: The work entitled ‘Total Phenolic Fraction (TPF) from Extra Virgin Olive Oil: induction of apoptotic-like cell death in Leishmania spp. promastigotes and in vivo potential of therapeutic immunomodulation’ aims to establish the antileishmanial activity of Extra Virgin Olive Oil. It is a very poorly executed study, the data is represented very poorly and cannot be considered for any scientific journal. The starting sentence of ‘Leishmaniasis being a disease unreliable chemotherapy’ is an incorrect statement. 

1. The assessment of promastigote growth by differential counting of dead and live promastigotes by using the Trypan blue exclusion dye is a very crude and inaccurate approach. In view of the CFSE staining method available, this Trypan blue approach is redundant. 

2. Line 210: Please state the incubation periods

3. A major problem in all assays is that it is stated that the concentration of the extract selected for this study is the IC50 and 2X1C50, and therefore the results section should begin with a viability assay that calculates this value. No where could I find a numerical value for the IC50 which is a major drawback. 

4. How was the cell size determined by Flow Cytometry? A reference is needed of this methodology. Composite results is not provided, only a representative profile.

5. In Fig. 2A and 2B, why is there no cell growth, cells appear to be in stationary phase. However, all experiments pertaining to cell growth should always be performed with log phase cells. If an IC50 conc. Decreases cell viability to 57%, it is expected that with 2X IC50, there should be a sharper fall in cell viability, or is the drug cytostatic?

6. In Fig 3A & 3B, Leishmania spp. promastigote proliferation determined by CFSE staining, there are no values stated, just a representative profile.

7. For the cell cycle study, it has been stated that ‘induced a significant increase in the number of parasites in the sub G0 peak region, consisting the 68.8% ± 8% for IC50 concentration and the 27.9% ± 4.1% 442 for 2 x IC50 concentration, compared to 10.4% ± 5.3% of untreated parasites (negative control group) (Fig 4A, 4B). Why should untreated promastigotes undergo apoptosis? Why should an 2XIC50 conc show a lesser degree of apoptosis than the IC50? The data is very poorly presented. 

8. In the Annexin V data, I fail to understand why do control cells have such a high % of Annexin V positivity, There are no values stated in the representative profiles.

The data needs to be thoroughly re-analysed and properly represented. It is written in a very rudimentary fashion.

Reviewer #2: No comments

PLOS authors have the option to publish the peer review history of their article (what does this mean?). If published, this will include your full peer review and any attached files.

Reviewer #1: No

Reviewer #2: Yes: Javier Moreno
---

## [Editor Report · Decision Letter 1]

9 Nov 2020

Dear Dr DOTSIKA,

We are pleased to inform you that your manuscript 'Total Phenolic Fraction (TPF) from Extra Virgin Olive Oil: induction of apoptotic-like cell death in Leishmania spp. promastigotes and in vivo potential of therapeutic immunomodulation.' has been provisionally accepted for publication in PLOS Neglected Tropical Diseases.

Best regards,

Hira L Nakhasi

Associate Editor

S Madison-Antenucci

Deputy Editor

The authors have revised the manuscript satisfactorily as per the reviewer's comments and have either reanalyzed the data or added additional information to support their conclusions.

---

## [Editor Report · Acceptance letter]

3 Dec 2020

Dear Dr DOTSIKA,

We are delighted to inform you that your manuscript, "Total Phenolic Fraction (TPF) from Extra Virgin Olive Oil: induction of apoptotic-like cell death in Leishmania spp. promastigotes and in vivo potential of therapeutic immunomodulation.," has been formally accepted for publication in PLOS Neglected Tropical Diseases.

Best regards,

Shaden Kamhawi

co-Editor-in-Chief

Paul Brindley

co-Editor-in-Chief
